# End-to-End Censored Demand Modeling

**Team Members: Natchapol Lebkrut (2026400198), Apithep Wongpiyachon (2026400200)**

**Course: Advanced Network Management（80240663-0）**

**Date: 2026.04.27**

**This proposal presents an end-to-end transformer framework for demand forecasting in perishable retail, unifying latent demand recovery with censorship-robust modeling. Having completed initial research and design, the project is now focused on implementation and validation using synthetic data to provide a scalable, accurate solution for supply chain optimization.**

# Table of Contents

# 1. Introduction

Forecasting demand for retail goods is difficult due to demand underestimation from censored sales data during stockouts. We propose the End-to-End Censorship-Aware Transformer (ECAT) framework, which unifies latent demand recovery and censorship-robust forecasting into a single differentiable model to handle the non-random nature of demand censoring.

### 1.1 Gaps Identified

Current models incorrectly assume missing data is random, failing to account for the non-random (MNAR) nature of stockouts. Additionally, existing methods often decouple imputation and forecasting, causing error propagation and preventing the models from leveraging shared predictive signals.

### 1.2 Novelty / Contribution

ECAT is the first end-to-end transformer to explicitly handle latent demand and forecasting. Its innovations include a differentiable architecture for joint learning, a censorship-aware block to model MNAR data, and a joint Tobit-Likelihood loss function to balance past reconstruction with future demand prediction.

# 2. Motivation

Inefficient supply chains for perishable goods lead to significant economic loss and food waste. Accurate demand forecasting is vital for inventory management, yet current models systematically ignore demand censoring during stockouts, treating zero sales as true zero demand. This creates a blind spot that undermines supply chain efficiency. We are developing a censorship-aware transformer pipeline to enable models to explicitly adapt to the non-random nature of stockouts in real time, moving toward proactive and accurate control of perishable food supply chains.

# 3. Background

Efficient supply chain management for perishable goods requires integrating traffic pattern forecasting with item-level demand prediction. This problem is complicated by non-random missing data patterns caused by stockout censorship.

Demand forecasting has evolved from statistical models (ARIMA) to deep learning (Transformers). Our project unifies high-capacity architectures with explicit MNAR censorship modeling using a Tobit-based end-to-end framework. This approach recovers "latent demand"—the true unobserved demand during stockouts—by optimizing all parameters simultaneously to prevent error propagation.

# 4. Related Works / Existing Methods

This section reviews literature relevant to demand forecasting and data censoring, categorized by methodological approach.

### 4.1 Classical Time-Series Forecasting

Classical statistical methods like ARIMA and exponential smoothing are effective for linear patterns but struggle with non-stationary data and complex dynamics especially for censored data.

### 4.2 Censoring Models

Econometric models like the Tobit model provide a statistical basis for estimating latent variables from truncated data. However, these are generally restricted to simple regression with strong distributional assumptions. Existing time-series adaptations typically use two-stage methods that lack end-to-end integration, leading to error propagation.

### 4.3 Time-Series Imputation

Deep learning techniques (e.g., GRU-D, SAITS) capture complex temporal patterns for imputation but predominantly focus on Missing At Random (MAR) scenarios. By treating stockouts as random omissions, they systematically underestimate demand. No existing method integrates these techniques with multi-horizon forecasting in an end-to-end MNAR pipeline.

## 5. Challenges

This project addresses data, modeling, and validation hurdles in censored demand forecasting.

### 5.1 Data Challenges

Retail data is sparse and noisy. Frequent stockouts and promotions create complex non-random missing patterns (MNAR), requiring high-fidelity simulation to generate realistic training data.

### 5.2 Modeling Challenges

Design challenges include selecting optimal time-series representations, simulating rare demand spikes, and balancing a complex joint Tobit-Likelihood loss function to prevent multi-stage error accumulation.

### 5.3 Validation Challenges

The lack of ground truth for latent demand during stockouts makes direct validation impossible. Success relies on using forecasting performance as a proxy to demonstrate improvements over SOTA baselines.

## 6. Objectives

We aim to build the ECAT framework to unify demand recovery and forecasting. Key goals include modeling stockouts as MNAR problems using a custom Tobit-Likelihood loss and survival functions.

We will develop a shared encoder to reduce error propagation, construct a high-fidelity simulation for synthetic data generation, and rigorously evaluate improvements in accuracy and bias over current methods.

# 7. Proposed Methodology

The ECAT framework uses a unified, end-to-end architecture to prevent error propagation. Raw time-series data, including historical sales and stockout indicators, are embedded with categorical features and combined with position encodings to provide temporal context. This sequence passes through a multi-layer shared transformer encoder to generate high-level feature representations.

A key innovation is the Censorship-Aware Transformer Block, which explicitly incorporates the observation indicator at every layer to model the MNAR missingness mechanism. The resulting shared output is processed by parallel probabilistic decoders: a Latent Demand Recovery Head for imputation during stockouts and a Censoring-Robust Demand Forecasting Head for future predictions. Optimization is achieved via a single custom Joint Loss Function that integrates Tobit-inspired survival functions to handle censorship constraints globally.

## 7.1 Overall Pipeline / Architecture

The ECAT is an end-to-end transformer unifying latent demand recovery and censorship-robust forecasting into a single differentiable model. It features a multi-layer shared transformer encoder whose output is processed by parallel probabilistic decoders and optimized via a single custom joint loss function.

## 7.2 Step 1: Data Collection and Preprocessing

Input data includes Observed Sales ($y_t$), the Stockout Indicator ($s_t$), Promotional Depth, Weather, and Calendar variables. Preprocessing involves standard loading, normalization, and embedding of categorical variables, with the explicit inclusion of $s_t$ making the model censorship-aware.

## 7.3 Step 2: Core Model Design

The core model is a Shared Multi-Head Transformer Encoder featuring a Variable Selection Network (VSN) with Gated Linear Units for explainability. Temporal Self-Attention captures complex time dependencies. The encoder outputs the high-dimensional Latent Demand Representation ($Z_t$).

## 7.4 Step 3: Training Strategy

The training utilizes two parallel probabilistic decoder heads: Latent Demand Recovery (for imputation) and Censoring-Robust Demand Forecasting (Multi-Horizon Temporal Fusion Decoder). All parameters are jointly optimized by a custom Joint Tobit-Likelihood Loss, applying Tobit logic during stockouts ($s_t=0$).

## 7.5 Step 4: Evaluation Plan

Evaluation involves an assessment focusing on accuracy and latent recovery. Since direct ground truth validation is impossible, success is evaluated by improved forecasting performance and reduced bias over SOTA baselines.

## 7.6 Data Simulation and Validation

A high-fidelity food chain simulation constructs a synthetic dataset with known ground truth for direct validation. This dataset, including the critical binary stockout indicator, is vital for calculating metrics like WAPE against the true latent demand.

## 8. Dataset

        The FreshRetailNet-50k dataset is used to be our benchmark on this problem since it has truly

## 9. Project Schedule

The remaining project timeline is broken down by week below.

| Week | Phase | Tasks | Detail | Lead / Support |
|---|---|---|---|---|
| 1 | Simulation Setup & Initial Training | Simulation Fine-tuning, ECAT Implementation, and Initial Training | Fine-tune high-fidelity simulation (5,000 combinations); Implement ECAT and ensure initial convergence of joint Tobit-Likelihood loss. | Apithep (Lead - Implementation, Support - Simulation) / Natchapol (Lead - Simulation, Support - Implementation) |
| 2 | Model Tuning & Baseline Comparison | Detailed Model Tuning and SOTA Baseline Comparison | Perform hyperparameter sweep (LR, layers); Benchmark against two-stage and MAR-assuming SOTA models. | Apithep (Lead - Tuning/Comparison, Support - Implementation) / Natchapol (Lead - Baselines, Support - Tuning) |
| 3 | Ablation Study & Detailed Evaluation | Ablation Study and Multi-Metric Evaluation | Quantify impact of Censorship-Aware block and measure bias reduction (WAPE) in latent demand recovery. | Natchapol (Lead - Ablation/Metrics) / Apithep (Support - Data Prep/Visualization) |
| 4 | Finalization & Report Writing | Final Analysis, Report Writing, and Presentation Prep | Synthesize results, finalize report structure, and draft presentation slides. | Natchapol (Lead - Analysis/Writing) / Apithep (Lead - Presentation/Final Review) |

## 10. Progress So Far

- **Completed:**
  - **Research & Design:** Finished the comprehensive literature review and fully designed the End-to-End Censorship-Aware Transformer (ECAT) framework, including the shared encoder, censorship-aware block, and custom joint Tobit-Likelihood loss function.
  - **Initial Setup:** Selected and configured the core simulation environment and formulated a validation plan using indirect metrics.
- **Ongoing:**
  - **Implementation & Training:** Implementing the initial ECAT model and conducting preliminary training to test convergence and computational intensity.
  - **Data & Evaluation Prep:** Fine-tuning the food chain simulation for high-fidelity MNAR scenarios and preparing state-of-the-art baseline models for rigorous comparative evaluation.

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

## Appendix I - FreshRetailNet-50K Dataset and Exploratory Data Analysis

This section shows the data characteristic and the distribution of the dataset by aggregation and visualization on the library pandas, matplotlib, and seaborn.

| Category | Data Columns Included |
|----------|----------------------|
| **Identifiers** | city_id, store_id, product_id, management_group_id |
| **Product Hierarchy** | first_category_id, second_category_id, third_category_id |
| **Sales / Time** | dt (Date), sale_amount, hours_sale (24h array) |
| **Censorship Indicators** | stock_hour6_22_cnt, hours_stock_status (24h binary array) |
| **External Features** | discount, holiday_flag, activity_flag, precpt, avg_temperature, avg_humidity, avg_wind_level |

```
Dataset Info:
<class 'pandas.core.frame.DataFrame'>
RangeIndex: 4500000 entries, 0 to 4499999
Data columns (total 19 columns):
 #   Column              Dtype
---  ------              -----
 0   city_id             int64
 1   store_id            int64
 2   management_group_id int64
 3   first_category_id   int64
 4   second_category_id  int64
 5   third_category_id   int64
 6   product_id          int64
 7   dt                  datetime64[ns]
 8   sale_amount         float64
 9   hours_sale          object
 10  stock_hour6_22_cnt  int32
 11  hours_stock_status  object
 12  discount            float64
 13  holiday_flag        int32
 14  activity_flag       int32
 15  precpt              float64
 16  avg_temperature     float64
 17  avg_humidity        float64
 18  avg_wind_level      float64
dtypes: datetime64[ns](1), float64(6), int32(3), int64(7), object(2)
```

(i) Columns and Data Types of the Dataset

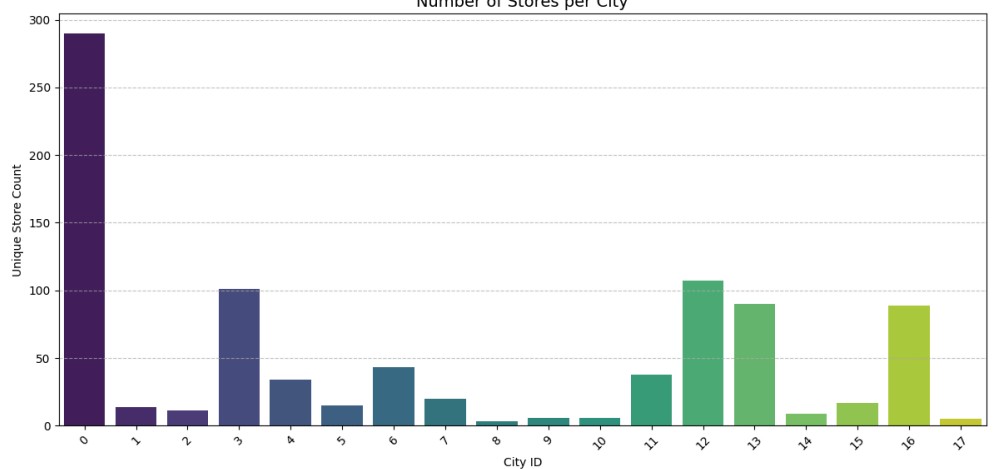

(ii) Distribution of the store in each city

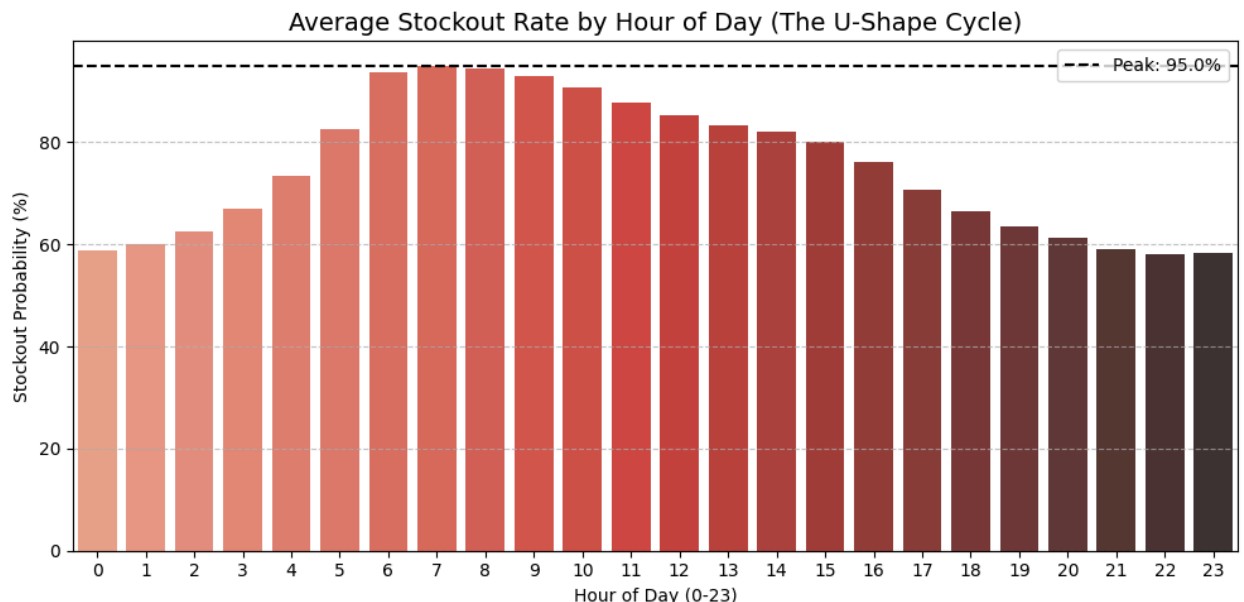

(iii) Example plot of the stockout distribution against time

iv) Example plot of the multimodal covariates against time

# Appendix II - Proposed Model Architecture End-to-End Censorship-Aware Transformer (ECAT)

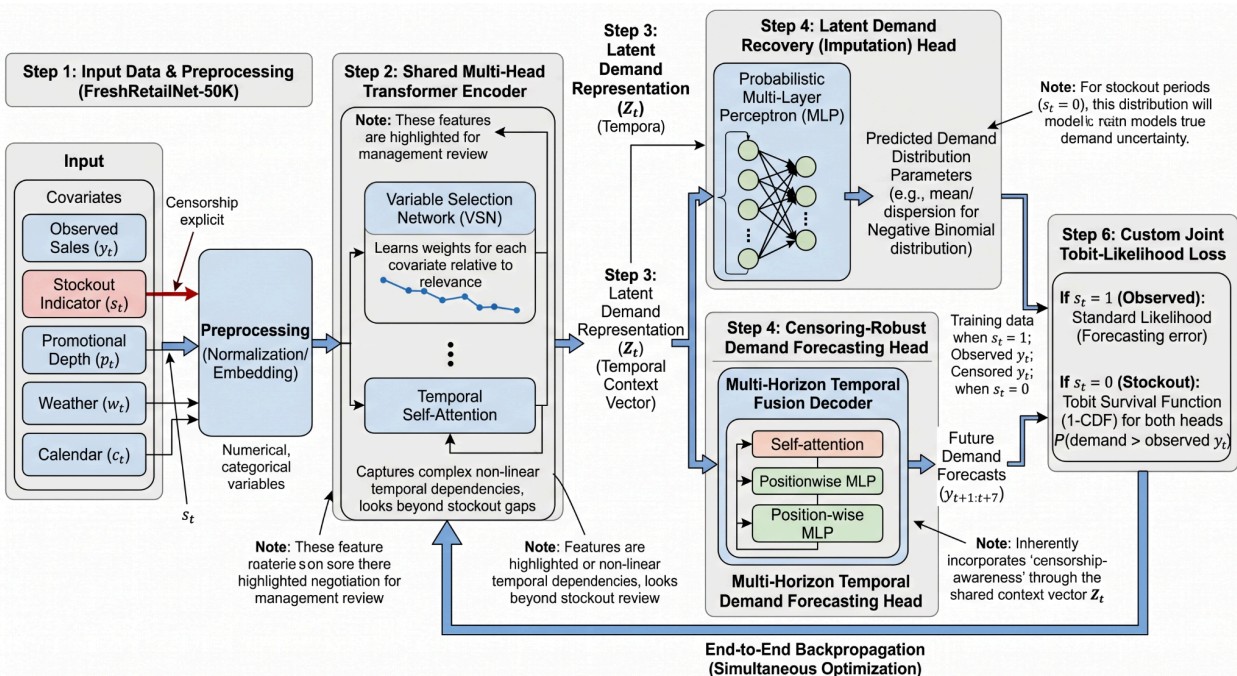

The End-to-End Censorship-Aware Transformer (ECAT) is designed to fix a major flaw in retail forecasting: the model assuming that "zero stock" means "zero demand." Instead of using a clunky two-step process (guessing missing data, then forecasting), ECAT does both at the same time using a shared "brain."

## Step 1: Inputs & Preprocessing

$$\mathbf{u}_t = [y_t \oplus s_t \oplus \mathbf{x}_t^{(num)} \oplus \text{Embed}(\mathbf{x}_t^{(cat)})]$$

We feed the model our daily sales, along with external factors like weather, holidays, and discounts. The most important input is the stockout indicator flag telling the model whether the item was in stock or sold out that day.

## Step 2: The Shared Encoder

Instead of separate models for filling in missing data and predicting the future, ECAT uses one shared Transformer encoder for these two methods:

- Variable Selection Network (VSN): This learns which features actually matter on a given day (e.g., heavily weighing the "discount" feature if there's a big sale).

$$v_t^{(j)} = \text{Softmax}(\mathbf{W}_{vsn}\mathbf{u}_t + \mathbf{b}_{vsn})_j$$

- Temporal Self-Attention: This looks back at historical data to spot complex trends, helping the model "remember" past demand even if recent days were out of stock.

$$\text{Attention}(\mathbf{Q}, \mathbf{K}, \mathbf{V}) = \text{Softmax}\left(\frac{\mathbf{Q}\mathbf{K}^T}{\sqrt{d_k}}\right)\mathbf{V}$$

**Step 3: The Context Vector**

The encoder compresses all that historical and contextual data into a single vector. This vector represents the "true" demand and gets passed to two separate heads simultaneously.

**Step 4: Recovery Head**

When the stockout flag says shelves were empty , this head steps in to guess what the demand should have been. Because we can't know for sure, it doesn't just guess a single number. Instead, it outputs a probability distribution, which gives us a range of uncertainty.

**Step 5: Forecasting Head**

Running in parallel, this head takes the exact same vector and predicts future sales for the next 7 days. Because it uses the shared context vector, its predictions aren't dragged down by past stockout data.

**Step 6: The Custom Loss Function**

$$\mathcal{L}_{rec,t} = -\left[s_t \log P_{\text{NB}}(y_t \mid \mu_t, \alpha_t) + (1 - s_t) \log\left(1 - F_{\text{NB}}(y_t - 1 \mid \mu_t, \alpha_t)\right)\right]$$

This is what makes the model "censorship-aware." The loss function dynamically changes how it grades the model based on the stockout flag:

- If in-stock: It calculates error normally, since the recorded sales equal the true demand.
- If out-of-stock: It uses "Tobit" logic. It knows the recorded sales were capped by inventory limits. So, it only penalizes the model if it guesses a demand *lower* than the recorded sales. It won't penalize the model for guessing higher.

$$\mathcal{L}_{total} = \sum_t \left(\text{MSE}(y_{t+k}, \hat{y}_{t+k}) + \lambda\mathcal{L}_{rec,t}\right)$$