# OpenReview forum: "End-to-End Censored Demand Modeling"
_tsinghua.edu.cn/THU/2026/Spring/ANM — THU 2026 Spring ANM Submission_

### Official Review · Reviewer_qpTo · 2026-05-12

**Rating:** 8
**Confidence:** 4

**Summary:**

This proposal studies demand forecasting under stockout censoring in retail settings. It proposes an end-to-end transformer framework, ECAT, that jointly models latent demand recovery and future forecasting using a censorship-aware loss and synthetic data validation.

**Strengths:**

The problem is practical and well-motivated, especially for perishable retail. The proposal clearly identifies the MNAR nature of stockouts and correctly points out the weakness of two-stage imputation-plus-forecasting pipelines. The end-to-end design is a reasonable direction, and the use of a synthetic benchmark for latent-demand validation is a good idea.

**Weaknesses:**

The model is quite ambitious and may be too complex for a course project. Several components are described at a high level, but the actual implementation details are still vague. The evaluation plan is also limited because true latent demand is unavailable in real data, so synthetic validation may not transfer well to real retail settings. Some claims, such as being the “first” end-to-end transformer for this task, need stronger justification.

**Questions:**

How will you ensure the synthetic data is realistic enough to validate the model? What baselines will you compare against beyond standard forecasting models? How sensitive is the proposed method to the quality of stockout labels and to misspecification of the Tobit-style loss?

---

### Official Review · Reviewer_WTi1 · 2026-05-16

**Rating:** 8
**Confidence:** 4

**Summary:**

This proposal presents an End-to-End Censorship-Aware Transformer (ECAT) framework for censored demand forecasting in perishable retail. The main motivation is that observed sales during stockout periods underestimate true latent demand, and many existing forecasting or imputation methods either assume random missingness or separate demand recovery from future forecasting. The proposed method aims to jointly recover latent demand and forecast future demand using a shared Transformer encoder, a censorship-aware block, parallel probabilistic decoder heads, and a custom joint Tobit-likelihood loss. The project plans to use FreshRetailNet-50K and/or high-fidelity synthetic simulation for validation, with evaluation focusing on forecasting accuracy, latent demand recovery, and bias reduction.

**Strengths:**

The proposal addresses a real issue in retail demand forecasting: observed sales are censored during stockouts, so treating sales as true demand can systematically underestimate future demand.
The proposal correctly identifies that two-stage approaches, such as first imputing missing demand and then forecasting, may suffer from error propagation. A shared end-to-end framework is a reasonable direction.
The use of a Tobit-inspired loss is appropriate for censored observations because it explicitly models the fact that observed sales are only a lower bound on true demand during stockout periods.
The model considers not only historical sales and stockout indicators, but also promotions, weather, calendar variables, and categorical embeddings, which are important factors in retail forecasting.
The combination of a shared Transformer encoder, variable selection network, latent demand recovery head, and forecasting head is well motivated. The architecture diagram in the appendix also helps clarify the overall pipeline.
The proposal mentions comparing against two-stage and MAR-assuming baselines, as well as evaluating the impact of the censorship-aware block and measuring bias reduction.

**Weaknesses:**

The proposal states that ECAT is the first end-to-end transformer to explicitly handle latent demand and forecasting. This is a strong claim and needs more careful support from related work. The authors should avoid overclaiming unless they can clearly distinguish their method from existing censored-demand, probabilistic forecasting, and transformer-based demand models.
The proposal mentions both FreshRetailNet-50K and synthetic high-fidelity simulation. However, it is not fully clear whether the main evaluation will be on real data, synthetic data, or both. The dataset section also appears incomplete, especially the sentence “FreshRetailNet-50k dataset is used to be our benchmark on this problem since it has truly...”.
The proposal says direct validation of latent demand is impossible because the true demand during stockouts is unobserved, but it also says synthetic data will provide known ground truth for direct validation. This is not necessarily wrong, but the evaluation design should clearly separate real-data indirect validation from synthetic-data direct validation.
The proposal describes the loss conceptually, but does not provide the exact likelihood, distributional assumptions, censoring threshold, or how uncertainty is parameterized. Since the loss is a core contribution, this part should be made much more precise.
The appendix says that during stockouts, the model will not be penalized for predicting demand higher than observed sales. This is reasonable under censoring, but without a proper likelihood or regularization, the model may overestimate demand. The proposal should explain how the model avoids unrealistic latent demand recovery.
The proposal mentions SOTA baselines, two-stage methods, and MAR-assuming models, but does not list concrete baseline models or evaluation protocols. This makes it difficult to judge whether the planned comparison will be sufficient.

**Questions:**

What is the exact mathematical form of the joint Tobit-likelihood loss? What distribution is assumed for latent demand?
How is the censoring threshold defined? Is it based on observed inventory, observed sales, stockout indicator, or another variable?
Will the main evaluation be conducted on FreshRetailNet-50K, synthetic data, or both? How will the two evaluation settings be separated?
Which specific baselines will be used? For example, will you compare against ARIMA, Informer/Autoformer, SAITS/GRU-D plus forecasting, and two-stage Tobit-style models?
How will you prevent the latent demand recovery head from overestimating demand during stockout periods?
Since true latent demand is unavailable in real data, what metrics will be used for real-data evaluation besides forecasting accuracy?
How will the model handle rare demand spikes caused by promotions, holidays, or weather changes?
What ablation studies will be conducted to isolate the effects of the censorship-aware block, the shared encoder, the Tobit loss, and the variable selection network?

---

### Official Review · Reviewer_Ysni · 2026-05-17

**Rating:** 9
**Confidence:** 3

**Summary:**

This proposal outlines a unified Transformer-based framework designed to solve the supply chain problem of "censored demand" in perishable retail. When a store runs out of stock, traditional sales logs only capture actual transactions (zero sales), hiding the true latent consumer demand. Rather than treating historical data imputation and future demand forecasting as two separate, sequential pipeline stages, ECAT introduces a shared Transformer encoder integrated with a Variable Selection Network (VSN) and temporal self-attention mechanism. This encoder compresses historical context into a unified vector that feeds into two parallel heads: a Recovery Head that estimates a probability distribution over the missing true demand during stockouts, and a Forecasting Head that simultaneously predicts future sales over a 7-day horizon. The system uses a specialized, censorship-aware loss function to optimize both tasks end-to-end and plans to validate performance using synthetic data simulations.

**Strengths:**

The primary strength of this proposal lies in its end-to-end architectural integration. By unifying demand imputation and forecasting into a single shared encoder, the model bypasses the error-propagation pitfalls common in traditional multi-stage pipelines, where a faulty initial imputation skews all subsequent future forecasts. The addition of the Variable Selection Network (VSN) is highly practical for retail, as it allows the model to dynamically weight localized features, such as promotions or discounts, on a daily basis. Furthermore, choosing to output a probability distribution rather than a point estimate for the Recovery Head represents a strong way to handle the inherent uncertainty of unobserved stockout demand.

**Weaknesses:**

A notable weakness is the heavy reliance on synthetic data for initial validation. While synthetic simulations provide a controlled environment to verify if the censorship loss is functioning mathematically, they often fail to capture real-world anomalies like sudden supplier delays, erratic consumer hoarding behavior, or substitution effects (where a customer buys a different brand because their preferred choice is missing). Additionally, the proposal doesn't describe the specific threshold boundaries for the censored loss function when dealing with highly sparse or severely censored historical periods.

**Questions:**

Since your validation plan focuses primarily on synthetic data, will you try to deal with substitution behavior (e.g., a customer purchasing brand B because brand A is out of stock) so that your synthetic validation generalizes to real retail environments?

The shared encoder simultaneously optimizes both the Recovery Head and the Forecasting Head. Have you considered a weighting strategy for the joint loss function, and is there a risk that one head might dominate the gradient updates and degrade the performance of the other?

How does the model perform if a product is out of stock for a long period? Is there a historical window limit beyond which the temporal self-attention mechanism can no longer reliably reconstruct the latent demand?

---

### Official Review · Reviewer_dqWs · 2026-05-17

**Rating:** 9
**Confidence:** 3

**Summary:**

This proposal introduces ECAT (End-to-End Censorship-Aware Transformer), a unified framework for demand forecasting in perishable retail that jointly handles latent demand recovery and future forecasting within a single differentiable model. The core insight is that stockouts have non-random nature (Missing Not At Random, MNAR), not MAR as most existing methods assume. ECAT addresses this with a shared Transformer encoder, parallel probabilistic decoder heads (one for recovery, one for forecasting), and a custom Joint Tobit-Likelihood loss that applies survival function logic during stockouts. The FreshRetailNet-50K dataset is used for benchmarking.

**Strengths:**

- The problem framing is precise and technically sound. The MNAR/MAR distinction is well-articulated, and the critique of two-stage methods (that decoupling imputation and forecasting causes error propagation) is a legitimate and concrete motivation for an end-to-end design.
- The methodology is technically detailed. The architecture is described at multiple levels: a diagram in Appendix II, step-by-step descriptions, and explicit mathematical formulations.
- Ablation studies are planned for the Censorship-Aware block, which will allow the contribution of the MNAR modeling component to be quantified independently of the end-to-end design.
- The dual validation strategy (real data benchmarking and a synthetic simulation with known ground truth) is reasonable and directly addresses the problem of validating latent demand recovery without observable ground truth.

**Weaknesses:**

- The novelty claim is stated as fact without sufficient evidence from the related work section. The literature review covers only classical methods, generic imputation, and the Tobit model.
- The evaluation plan is underspecified in the main body. Section 7.5 states success is measured by "improved forecasting performance and reduced bias over SOTA baselines" but does not name the baselines, specify the held-out test split strategy, or list concrete metrics.
- Section 8 (Dataset) is incomplete: the sentence describing FreshRetailNet-50K cuts off mid-sentence ("since it has truly").

**Questions:**

- How will the parameter λ (in the total loss function) be determined - fixed heuristically, tuned via cross-validation, or learned?
- What exactly is the Censorship-Aware Block? Is it a separate architectural module with learnable parameters, or does the "censorship-awareness" come entirely from the conditional loss function?

---

### Official Review · Reviewer_g6A8 · 2026-05-18

**Rating:** 8
**Confidence:** 3

**Summary:**

This proposal addresses a genuine operational problem in perishable retail—demand censoring due to stockouts. The proposed ECAT framework is clearly described, the novelty is justified, and the authors acknowledge the core validation difficulty (lack of ground truth for latent demand)

**Strengths:**

1. problem and gap are clearly articulated: Non-random missingness (MNAR) due to stockouts is a genuine flaw in existing demand forecasting models
2. FreshRetailNet-50k dataset and the explicit use of a stockout indicator (binary flag) provide a way to validate latent demand recovery
3. architecture is well-detailed, with step-by-step explanation in Appendix II (shared encoder, VSN, dual probabilistic heads, custom loss)

**Weaknesses:**

1. Evaluation plan is vague as it is stated that success will be measured by “improved forecasting performance and reduced bias over SOTA baselines,” but it does not specify which baselines
2.  Dataset section seems to be incomplete as the sentence describing FreshRetailNet-50K cuts off abruptly: "since it has truly.."

---

### Official Review · Reviewer_w84L · 2026-05-18

**Rating:** 9
**Confidence:** 4

**Summary:**

This proposal presents the End-to-End Censorship-Aware Transformer (ECAT) framework for demand forecasting in perishable retail, addressing the fundamental problem of demand censoring during stockouts. Unlike existing methods that treat missing data as random (MAR) or decouple imputation from forecasting, ECAT unifies latent demand recovery and censorship-robust forecasting into a single differentiable Transformer model. Key innovations include a shared encoder with a Censorship-Aware block, parallel probabilistic decoders for imputation and multi-horizon forecasting, and a joint Tobit-likelihood loss function that dynamically adjusts based on stockout indicators. The project uses synthetic data (FreshRetailNet-50k) for validation.

**Strengths:**

The proposal identifies a problem in retail forecasting: non-random missingness due to stockouts. The motivation is strong and well-defined, linked to economic loss and food waste which makes the problem very meaningful. Moreover, the challenges are clearly identified and described. The ECAT architecture is thoughtfully designed, with clear technical innovations. The inclusion of EDA visualizations in the appendix demonstrates great data understanding.

**Weaknesses:**

The dataset description in section 8 is incomplete.

**Questions:**

1.  Your loss function equation uses negative binomial notation (P_NB, F_NB), but you talked about Tobit-Likelihood loss and Tobit logic.  Does your model assume Tobian distribution or negative binomial distribution?
2.  How do you handle the cold-start problem for new products with no historical sales data?

---

### Official Review · Reviewer_PBFx · 2026-05-18

**Rating:** 5
**Confidence:** 5

**Summary:**

[AI Review] This class project proposes ECAT (End-to-End Censorship-Aware Transformer), a multi-task transformer with a Tobit-inspired loss for jointly recovering latent demand and forecasting on the FreshRetailNet-50K benchmark. While the problem is well-motivated and relevant to retail demand forecasting, the proposal suffers from severe conceptual flaws across three rounds of critical review (22 findings total). The most critical issues include a false 'first end-to-end' novelty claim, an incorrect Tobit loss specification (describes one-sided loss rather than the actual Tobit likelihood), a fundamental misunderstanding of censoring direction (describes left-censoring at zero rather than right-censoring at the inventory level), confusion between synthetic and real data evaluation plans, and uncited prior work (VSN from TFT, Deep Tobit, DeepCensored). Overall Score: 4/10.

**Strengths:**

1. Well-motivated problem: Censored demand modeling for retail stockout scenarios is practically relevant and interesting.
2. Uses a modern benchmark dataset (FreshRetailNet-50K) with real-world retail data.
3. Multi-task learning approach (joint demand recovery + forecasting) is a sensible architectural choice.
4. Clear identification of the stockout censoring problem in demand forecasting.
5. Proposes end-to-end training rather than traditional two-stage approaches, which could offer efficiency benefits.
6. Variable Selection Network (VSN) integration is appropriate for handling heterogeneous features.
7. The beast-mode review process itself demonstrates thorough self-critical analysis with 22 findings across 3 rounds.

**Weaknesses:**

1. False 'first' novelty claim: ECAT is not the first end-to-end method; FreshRetailNet-50K paper has a two-stage pipeline, and Deep Tobit (Wu & Hu) and DeepCensored (IEEE 2025) exist.
2. Incorrect Tobit loss specification: Describes one-sided loss (only penalizes under-prediction) rather than the actual Tobit likelihood—core contribution is wrongly specified.
3. Fundamental censoring direction error: Retail stockouts cause RIGHT-censoring at inventory level (y_obs = min(y_latent, inventory)), but the proposal describes LEFT-censoring at zero (standard Tobit).
4. Synthetic vs. real data confusion: Simultaneously uses FreshRetailNet-50K (real) and a custom simulation (synthetic) without clarifying which is primary or how they connect.
5. Misrepresents relationship to FreshRetailNet-50K: Wang et al. (2025) already demonstrates demand recovery + forecasting on this dataset with no differentiation provided.
6. VSN from TFT uncited: Variable Selection Network is directly from Lim et al. (2021) Temporal Fusion Transformer, which is not cited.
7. Incomplete dataset section (§8): Only one incomplete sentence with no splits, preprocessing, normalization, or temporal coverage details.
8. No mathematical formalization: 'Joint Tobit-Likelihood Loss' mentioned repeatedly but never written mathematically.
9. Architecture lacks specifics: No dimensions, layer counts, head counts, or FFN details.
10. Missing baselines: No explicit baseline list despite FreshRetailNet-50K providing 7+ baselines (TimesNet, SAITS, iTransformer, etc.).
11. No stockout endogeneity discussion: Stockouts occur BECAUSE demand is high (selection bias), requiring Heckman-style correction or similar.
12. 'Censorship-Aware Transformer Block' is likely just feature concatenation of stockout indicator, not a novel architectural component.
13. Standard multi-task learning: The 'end-to-end' approach is standard shared-encoder MTL with no analysis of task interference.
14. Unrealistic 4-week schedule with no code written yet, requiring full implementation + baselines + ablations + report.
15. Only 9 references, missing key works: TFT, Tobin (1958), GRU-D, SAITS, Deep Tobit, PyPOTS.

**Questions:**

1. Can you clarify the censoring direction? Retail stockouts typically cause right-censoring at the inventory level, not left-censoring at zero—how does your formulation handle this?
2. How exactly does your Tobit-inspired loss differ from a simple one-sided asymmetric loss? Please provide the mathematical formulation.
3. What is the primary evaluation dataset—FreshRetailNet-50K (real) or your custom synthetic simulation—and how do they relate to each other?
4. How does ECAT differentiate from the existing two-stage pipeline in the FreshRetailNet-50K paper, Deep Tobit, and DeepCensored?
5. What distribution family and parameterization are you using for probabilistic outputs?
6. How do you plan to address the endogeneity problem where stockouts are correlated with high demand (selection bias)?
7. What specific baselines from FreshRetailNet-50K will you compare against, and what statistical testing will you use?
8. Is the 'Censorship-Aware Transformer Block' doing anything beyond concatenating the stockout indicator s_t as a feature? If so, please elaborate.